# Inflammation in Urological Malignancies: The Silent Killer

**DOI:** 10.3390/ijms24010866

**Published:** 2023-01-03

**Authors:** Martina Catalano, Giandomenico Roviello, Raffaella Santi, Donata Villari, Pietro Spatafora, Ilaria Camilla Galli, Francesco Sessa, Francesco Lupo Conte, Enrico Mini, Tommaso Cai, Gabriella Nesi

**Affiliations:** 1School of Human Health Sciences, University of Florence, 50134 Florence, Italy; 2Section of Clinical Pharmacology and Oncology, Department of Health Sciences, University of Florence, 50139 Florence, Italy; 3Section of Pathological Anatomy, Department of Health Sciences, University of Florence, 50139 Florence, Italy; 4Department of Experimental and Clinical Medicine, University of Florence, 50134 Florence, Italy; 5Unit of Urological Robotic Surgery and Renal Transplantation, Careggi Teaching Hospital, 50134 Florence, Italy; 6Histopathology and Molecular Diagnostics, Careggi Teaching Hospital, 50139 Florence, Italy; 7Department of Urology, Santa Chiara Regional Hospital, 38122 Trento, Italy

**Keywords:** inflammation, tumorigenesis, cancer, genitourinary tumors

## Abstract

Several studies have investigated the role of inflammation in promoting tumorigenesis and cancer progression. Neoplastic as well as surrounding stromal and inflammatory cells engage in well-orchestrated reciprocal interactions to establish an inflammatory tumor microenvironment. The tumor-associated inflammatory tissue is highly plastic, capable of continuously modifying its phenotypic and functional characteristics. Accumulating evidence suggests that chronic inflammation plays a critical role in the development of urological cancers. Here, we review the origins of inflammation in urothelial, prostatic, renal, testicular, and penile cancers, focusing on the mechanisms that drive tumor initiation, growth, progression, and metastasis. We also discuss how tumor-associated inflammatory tissue may be a diagnostic marker of clinically significant tumor progression risk and the target for future anti-cancer therapies.

## 1. Introduction

Although inflammation is a self-limiting host defense strategy against biological, chemical, and physical agents, it is also considered a hallmark of cancer development. Inflammation, especially if persistent, stimulates cell proliferation and local host response, causing cell damage and the development of various diseases including cancer [1,2]. In addition, the tumor microenvironment (TME), enriched in cytokines, chemokines, transcription factors, and immune cells, can enhance tumor growth and immune escape. Inflammation is an important participant in the narrative regarding tumors, and oncogenesis is estimated to correlate with chronic infection and inflammation in 15–20% of cancers [3]. Etiological factors such as Helicobacter pylori, hepatitis B or C, and autoimmune diseases, are linked to gastric and colorectal cancer, hepatocellular carcinoma, and mucosa-associated lymphoid tissue lymphoma [4,5].

Over the last few years, numerous studies have shown that inflammatory molecules and pathways promote the development of various cancer types, including genitourinary tumors [6,7,8]. In bladder cancer (BC), chronic inflammation (e.g., urogenital schistosomiasis) is accepted as a risk factor together with other well-established causes such as smoking and occupational exposure to aromatic amines [9]. Advanced age, African descent, family history, and genetic mutations are known to be risk factors for prostate cancer (PCa), but the mechanisms responsible for initiation and progression have not yet been fully elucidated [10,11]. Inflammation-induced cellular stress and repeated genomic damage have been investigated as possible causes of PCa [12,13]. Accumulating evidence shows that immune cells and inflammatory pathways can promote renal cell carcinoma (RCC) growth and immune escape [14], whereas there are insufficient grounds to claim a role for inflammation in other tumors of the urogenital tract, such as testicular and penile cancer.

In this review, we evaluate how inflammatory molecules trigger cell transformation, tumor cell proliferation, and the metastasis of genitourinary tumors, by analyzing the various signaling pathways and downstream transcription factors. We also describe the capacity of tumor-associated inflammatory cells to trigger tumor immune evasion and discuss future perspectives regarding the use of innovative anti-inflammatory agents and the mechanisms behind their anticancer activity.

## 2. Inflammation as Risk Factor in Genitourinary Cancers

Inflammatory response appears to be fundamental in the occurrence and development of tumors, however, its role in carcinogenesis still requires further clarification [12]. Several factors, such as pathogens, diet, mechanical and chemical trauma, are able to initiate an inflammatory process. Two different models have been proposed to describe how inflammation is related to cancer: (1) an intrinsic pathway induced by DNA damage, chromosomal instability, and epigenetic changes; and (2) an extrinsic pathway associated with inflammatory signals caused by autoimmune diseases or infections [15]. Both these pathways are characterized by the activation of transcription factors, such as Nuclear Factor-κB (NF-κB) and Signal Transducer and Activator of Transcription (STAT)-3, which drive the inflammatory cascade [16,17]. Overall, the most frequent causes of genitourinary inflammation reside in infectious and noninfectious etiology (Figure 1).

### 2.1. Infectious Agents

#### 2.1.1. Parasites

Schistosomiasis is a recognized risk factor for BC, the most common cancer type in men and the second in women in endemic regions of Sudan, Egypt, sub-Saharan Africa, and Yemen [18]. Squamous cell carcinomas constitute 60–90% and adenocarcinomas 5–15% of all schistosomiasis-associated BC, the rest being urothelial carcinomas [19]. Schistosoma haematobium eggs stimulate an inflammatory response often generating genotoxic factors that determine genomic instability and tissue damage. In vivo studies in CD-1 mice have shown that intravesical instillation of Schistosoma haematobium antigens causes inflammation and urothelial dysplasia [20]. It has been hypothesized that Schistosoma haematobium is oncogenic by inducing *K-RAS* mutations [21]. To combat helminthic infection, inflammatory cells release reactive oxygen species (ROS) and reactive nitrogen species which, chronically released at low concentrations, function as intra- and intercellular second messengers regulating cell proliferation, apoptosis, and gene expression [22,23].

#### 2.1.2. Bacterial Infections

The correlation between bacterial infections and the development of BC is still controversial. Some authors have reported a greater risk of BC in patients with a history of recurrent (three or more within 12 months) urinary tract infections (UTIs), whereas fewer UTI events treated with antibiotics correlate to a lower risk of BC [24,25]. Conversely, Jiang et al. demonstrated a significantly reduced risk of BC in patients with recurrent UTIs, possibly explained by the anti-cancer effect of the antimicrobial treatment, higher exposure to non-steroidal anti-inflammatory drugs, and immune response induced by bladder infection [26].

Gram-negative uropathogens are the main pathogens involved in bacterial prostatitis, however, Gram-positive and atypical microorganisms can also be causative agents [27]. Bacterial protein toxins could function as carcinogenetic stimuli, damaging DNA directly via enzymatic attack, indirectly by stimulating an inflammatory reaction that generates free radicals, or else by interfering with DNA repair mechanisms [28,29].

#### 2.1.3. Viruses

High-risk human papillomavirus (HPV) infection is a key factor in the development of several tumors, including penile carcinoma [30]. Cells harboring HPV genomes display increased levels of nitric oxide (NO), which fuels inflammation and enhances DNA damage [31]. The HPV genome contains genes encoding for E7 oncoprotein, which inactivates the retinoblastoma (Rb) tumor suppressor gene normally suppressing p16, and consequently high-level p16 expression is useful as a surrogate marker for HPV infection.

A thorough understanding of the role of HPV in bladder carcinogenesis has not been reached. Some authors have identified HPV as a causal factor [32,33] but this has not been confirmed by others [34,35].

HPV infection as a potential cause of PCa is a matter of debate. Since the prevalence of high-risk HPV in benign prostate tissue is similar to that observed in PCa samples, it could be sustained that the presence of HPV is coincidental. However, a systematic review by Lawson and Glenn has lately established that high-risk HPV types are more common in PCa than in benign tissue [36]. This finding is confirmed by a recent meta-analysis suggesting that HPV infection is related to an increased likelihood of PCa development, the odds ratio being 2.27 (95% CI, 1.40–3.69) [37].

Regarding hepatitis C virus (HCV) infection, recent data show a correlation between HCV infection and a greater risk of RCC. Conversely, this correlation is not statistically significant for PCa and BC [38].

Several authors have discussed the role of specific viruses in the development of testicular germ cell tumors (TGCTs). While high-quality evidence indicates that human immunodeficiency virus (HIV) and EBV infection increase the risk of TGCTs, results on the relationship with cytomegalovirus (CMV) and HPV are inconsistent, calling for further studies to draw firm conclusions [39].

### 2.2. Noninfectious Causes

The inflammatory process in the urogenital tract can be prompted by causes other than infectious. Local mechanical disorders may provoke metaplastic changes of the urothelium (e.g., cystitis cystica et glandularis, intestinal metaplasia, and nephrogenic adenoma) or lesions that carry a risk of progression to malignancy (e.g., keratinizing squamous metaplasia) [40].

Spinal cord injury causing neurogenic bladder and chronic indwelling catheter usage have been identified as risk factors for BC, but literature data are conflicting. Although a higher risk of BC has been reported in patients with spinal cord injury than in the general population [41,42], other evidence shows that the risk of squamous cell carcinoma (SCC) of the urinary bladder increases in patients with spinal cord injury requiring indwelling catheters (42%) rather than other types of catheterization (e.g., clean intermittent, condoms) or spontaneous urination [43]. Some authors have shown that, after adjustment for sex, age and comorbidities, any difference in BC incidence between patients with spinal cord injury and controls is negligible [44].

Regarding prostate inflammation, much attention has been focused on the role of diet, smoking, changes in serum testosterone and estrogen levels, autoimmunity, reflux of noxious chemicals in the urine, and metabolic alterations [8]. A high-fat diet correlates with a significant increase in pro-inflammatory cytokines and activation of pathways (e.g., STAT3 and NF-κB) involved in the proliferation, survival, angiogenesis, and invasion of prostate neoplastic cells [45]. In addition, studies in animal models have determined that estrogens can modulate the immune response and spur prostatitis [46,47]. Rats with both spontaneous and sex hormone-induced prostatitis display a reduced proportion of intraprostatic NK-like cells and a low ratio of NKT/T cells [48]. Significantly higher rates of proinflammatory cytokines, such as tumor necrosis factor (TNF)-α and macrophage inflammatory protein (MIP)-1α, have been found in estrogen-treated rats than in normal and castrated controls [49]. Despite this, the exact mechanisms of estrogen-mediated immunomodulation remain unclear.

## 3. Inflammation-Related Pathways in Genitourinary Cancers

Inflammation and genitourinary cancers are tightly connected, and numerous inflammatory pathways, including the vascular endothelial growth factor (VEGF), von Hippel-Lindau (VHL), the mechanistic target of rapamycin (mTOR), TNF and STAT signaling cascade, are involved.

### 3.1. VEGF Pathway

The VEGF is a signal protein implicated in the formation of blood vessels. The binding of VEGF to its receptors determines enhanced vascular permeability, increased cell growth, and the mobilization of bone marrow-derived endothelial progenitor cells for neovascularization. Vascular hyperpermeability induced by VEGF leads to loss of macromolecules in the extravascular space and the subsequent formation of edema, facilitating angiogenesis [50]. Through the PI3K/AKT pathway, VEGF also promotes the release of anti-apoptotic proteins such as BCL-2 and A1, favoring endothelial cell survival. With the release of proteases, mitogens, and adhesion molecules, it drives morphological changes in endothelial cells [51].

### 3.2. VHL Pathway

*VHL* gene encodes a protein implicated in the ubiquitination and degradation of hypoxia-inducible-factor (HIF), a transcription factor that has a vital role in the regulation of gene expression by oxygen levels. This protein also contributes to other cellular processes including cilia formation, cytokine signaling, regulation of senescence, and formation of the extracellular matrix [52]. In kidney cancer, molecular alterations in the *VHL* gene, with subsequent upregulation of hypoxia-inducible factor alpha (HIF-α), are pivotal in angiogenesis, proliferation, apoptosis, and metastasis. Under normal oxygen conditions, the HIF-α subunit is hydroxylated by specific prolyl-hydroxylases and targeted for rapid proteasomal degradation by the VHL tumor suppressor protein (pVHL). In a hypoxic environment or whenever functional pVHL is lacking, HIF-α is not degraded and translocates to the nucleus, where it dimerizes with HIF-β converting into transcriptionally active HIF. Moreover, HIF-independent gene expression changes induced by VHL loss may cooperate with HIF dysregulation in *VHL* defective tumorigenesis [53].

*VHL* silencing engenders the production of several proteins involved in angiogenesis (VEGF, phosphoglycerate kinase, glucose transport, and erythropoietin) and stimulation of chemokine receptors such as CXCR4, thus favoring metastatic spread [54]. Moreover, altered *VHL* seems to interfere with the structure of the kidney primary cilium, leading to renal cyst formation and possible cancer development [55].

Hypoxia also prompts RCC cells to secrete interleukin-6 (IL-6) and interleukin-8 (IL-8) potentially enhancing their invasive properties [56]. The AMP-activated protein kinase (AMPK) is a negative key regulator of IL-6 and IL-8 production, and its activation appears to decrease RCC cell invasion in vitro and in ex vivo RCC cells from a human clear cell carcinoma [56].

### 3.3. mTOR Pathway

Protein kinase B (AKT) and mTOR are central in numerous cellular functions, such as proliferation, adhesion, migration, invasion, and metabolism. While the mTORC1 signaling complex stimulates cell growth, the mTORC2 regulates cell polarity and cytoskeletal rearrangement [57]. Furthermore, mTOR determines the accumulation of HIF via downstream proteins S6K1 and eIF-4E, which prove to be more active in high-grade tumors with poor prognostic features [58]. Specifically, S6K1 is an important downstream effector of mTOR that promotes protein translation of key metabolic transcription factors such as HIF-1α [59]. Some studies have reported that the translation of HIF-1α is partly mediated by the regulation of eIF-4E free levels through the Akt/mTOR and ERK pathways. The phosphorylation of eIF-4E at Ser 209 increases affinity for mRNA cap-binding, and may also facilitate its entry into initiation complexes, thus regulating HIF-1α protein synthesis [60].

The PI3K/AKT/mTOR signaling pathway is activated by the synthetized chemokines. In PCa, its activation, mediated by the CXCR10 and CXCR7 complex, facilitates immune evasion, whereas the CXCL13/CXCR5 axis prompts proliferation, invasion and progression [61].

The upregulation of PI3K/AKT/mTOR signaling pathway has a prognostic value in penile SCC [62]. Indeed, increased PI3K/AKT/mTOR expression has been reported to be linked to lower recurrence risk and overall mortality in these patients, suggesting a possible role for the mTOR pathway biomarkers in penile SCC patients’ stratification [62].

### 3.4. TNF Pathway

The TNF signaling cascade is essential to many physiological and pathological processes, such as cell proliferation, differentiation, apoptosis, modulation of host immune response, and the induction of inflammation. TNF is a unique cytokine with two distinct actions depending on its two receptors, i.e., tumor necrosis factor receptor 1 (TNFR1) that elicits inflammation and tissue apoptosis, and TNFR2 that regulates the immune system and tissue regeneration [63]. In RCC, elevated tissue and plasma TNFR2 levels are significantly related to a higher grade of malignancy, and the strategy to reduce or block TNFR2 expression may effectively inhibit tumor progression [63,64]. Furthermore, TNF can induce epithelial-mesenchymal transformation (EMT) and promote RCC tumorigenesis, as well as direct cell invasion and migration by suppressing E-cadherin, upregulating vimentin and stimulating matrix metalloproteinase 9 (MMP-9) production [65,66].

### 3.5. STAT Pathway

The STAT pathway comprises seven proteins involved in both cytoplasmic signaling and nuclear transcription processes. They are activated by immunomodulatory and inflammatory cytokines, as well as growth factors such as eritropoietin, to transmit cytoplasmic signals via membrane receptor-associated Janus kinases (JAK) [67]. Persistent cytokine and growth factor signaling leads to accumulation of STATs as shown in several types of cancer [68]. The constitutive activation of STAT3 induces the expression of anti-apoptotic proteins such as BCL-XL and of C-MYC, which favor cancer cell survival and prevent terminal differentiation [69]. Blocking the STAT3 signaling cascade can inhibit expression of these proteins and render cells more liable to apoptosis [70].

In BC, STAT3 silencing by small interfering RNA vectors appears to hinder T24 cell proliferation in vitro and in vivo [71]. In tumor tissue from BC patients, NF-kB, STAT3 and their target genes (i.e., cyclin-D1, VEGF-A, and TGFβ1) have been found to be activated in comparison to healthy tissue, suggesting that chronic inflammation can promote tumor development [72].

### 3.6. Nuclear Factor-Kappa B Pathway

The transcription factor NF-κB is an essential mediator of inflammatory response and, being constitutively expressed in various tumor types, it is held to tie inflammation to cancer. It is composed of five proteins whose activity is induced by ROS, TNF-α, IL-1β, bacterial lipopolysaccharides, and isoproterenol among others [73]. NF-κB induces cytokines, adhesion molecules and angiogenic factors to be expressed at the tumor site, increasing survival and proliferation of neoplastic cells via regulation of genes related to cell-cycle control (e.g., cyclin-D1, c-MYC) and apoptosis (e.g., BCL-2, cFLIP) [74]. Prominent nodes of crosstalk are mediated by other transcription factors including STAT3, beta-catenin and p53. NF-κB expression has been related to histological grade and T category in urothelial carcinoma [75], although an updated meta-analysis does not confirm any association between NF-κB polymorphisms and an increased risk of BC [76].

The NF-κB signaling pathway is frequently activated in human PCa. Through the CXCL13-CXCR5 axis, NF-κB regulates cell migration and contribute to the invasive phenotype of PCa cells [77]. In murine models, the CXCL13-CXCR5 pathway has been related to the loss of phosphatase and tensin homologue (PTEN) implicated in CXCL8-CXCR1/2 upregulation [78]. In PTEN-deficient prostate tumors, CXCL8 enables secretion of CXCL2 and CXCL12 by stromal cells, thus supporting proliferation, invasion, and NF-κB dependent survival [79].

### 3.7. Others

A plethora of other signal transduction pathways boost cancer cell proliferation. These include NOTCH, insulin-like growth factor (IGF), and hedgehog pathways.

NOTCH receptor signaling is mediated through the transcription of target genes (e.g., cyclin-D1, p21CIP1, NF-kB, and c-MYC). In PCa, evidence suggests a tumor suppressive as well as an oncogenic role for NOTCH, depending on which downstream proteins are activated. The NOTCH/HES1 axis with consequent PTEN suppression is crucial to PCa progression, while NOTCH upregulation results in increased cancer aggressiveness and docetaxel resistance [80].

The binding of insulin and IGF-1 to their receptors determines activation of the PI3K/AKT pathway and loss of PTEN, which elicits PCa cell proliferation [81]. In PCa cell lines, IGF-1 appears to directly activate the androgen receptor in the absence of androgens, contributing to the failure of androgen deprivation therapy and development of castration-resistant prostate cancer (CRPC). A phase II study tested figitumumab (CP-751,871), a human anti IGF-1R monoclonal antibody, administered with docetaxel/prednisone in chemotherapy-naïve (arm A) and chemotherapy refractory (arm B) CRPC patients. Figitumumab significantly reduced prostate-specific antigen (PSA) levels in arm B [81]. Another human anti IGF-1R antibody, cixutumumab (IMC-A12/LY3012217), was investigated in a phase I/II trial in combination with temsirolimus in metastatic PCa patients, but showed limited antitumor activity and a high incidence of adverse events including hyperglycemia (100%), oral mucositis (63%), diarrhea (44%), and pneumonitis (44%) [82].

High levels of activated ERK, AKT and STAT3 have been observed in seminomas and nonseminomatous germ cell tumors [83], and may participate in tumor proliferation and survival. These signaling molecules could be activated by multiple receptors or be constitutively active within TGCTs, possibly explaining the weak correlations observed between chemokines such as CXCR4 and downstream effectors in germ cell tumors [84].

## 4. Inflammation-Related Molecules in Genitourinary Cancers

### 4.1. Cytokines

Cytokines are a class of small proteins that act as signaling molecules in the regulation of inflammation and modulation of cellular activities, including growth, survival, and differentiation [85]. Transforming growth factor (TGF) β1 is a cytokine that strongly influences cell growth and phenotype, and is overexpressed in several types of cancer.

Most RCC cell lines are resistant to TGF-β1-mediated growth suppression, which may promote transformation and/or progression of human RCC [86]. Studies on RCC with mutated VHL have shown that proinflammatory cytokines (e.g., IL-6, TNF-α, IL-1) and MMP-2 foster tumor spread and are overexpressed in more invasive cell lines [65]. Through the PI3K/AKT pathway and subsequent inhibition of GSK-3β, TNF-α has been found to facilitate migration of RCC cell lines [86].

IL-6 seems to be implicated in the regulation of BC metastatic spread via the STAT3 signaling pathway, with contradictory results. While some in vitro and in vivo studies demonstrate a reduction in IL-6-associated cell proliferation, migration and invasion [87], others show elevated IL-6 to be associated with more advanced clinical stage (T2-T4 vs. T1 and CIS), higher recurrence rate following treatment, and shorter survival [88]. The correlation between IL-8 and urothelial BC risk was assessed in a case-control study and found to be significantly higher in BC tissue samples than in healthy bladder mucosa [89,90]. IL-8 engages in the metastatic cascade by increasing angiogenic activity and inducing MMP-9 expression [91]. High-grade tumors display significantly higher MMP-9 and IL-8 expression levels than low-grade tumors [92]. Increased MMP-9 and IL-8 expression also tends to occur in pT1-T2 tumors in comparison to non-invasive tumors. In human BC cells, TNF-α causes the secretion of MMP-9, which accelerates tumor invasion and metastasis [93]. In addition, TNF-α overexpression has been correlated to angiogenesis in recurrent large bladder carcinomas [94].

### 4.2. Chemokines

Chemokines are proinflammatory cytokines that exert a key role in tumor growth and metastasis [86].

The elevated levels of the chemokine CXCL12 are frequently detected in RCC. CXCL12 binds to CXCR4 that is highly expressed following the loss of VHL function [54,95]. The CXCL12/CXCR4 axis induce a cascade of signals that internalizes the complex in the nucleus, favoring cellular growth, migration, and invasion [96]. Specifically, the receptor in complex with the Gαi subunit G protein leads to the inhibition of adenylyl cyclase–mediated cyclic adenosine monophosphate production and the mobilization of intracellular calcium. The dissociation of the Gαi subunit results in the activation of multiple downstream targets, including ERK1/2, MAPK, AKT, and JNK effectors. This complex stimulates chemotaxis through cytoskeletal rearrangements, actin polymerization, polarization, pseudopodia formation, and integrin-dependent adhesion to endothelial cells [97]. Tissues that constitutively express CXCL12 are the main sites of metastasis for CXCR4-expressing cancer cells. This chemotactic property is integral to RCC cell migration, though CXCR4 activation seems to occur at an earlier stage in RCC tumorigenesis, affecting cell cycle regulation, breach of tissue barriers, and apoptosis inhibition [98]. Furthermore, CXCR3 and its ligand CXCL11, known to be potent eosinophil chemotactic factors, facilitate proliferative responses of RCC cells by triggering a series of signal transduction events that involve transient release of intracellular calcium and cytoskeletal rearrangements [99].

CXCL8 has a major role in PCa proliferation and progression, as it promotes phosphorylation and signaling via Src kinase, focal adhesion kinase, PI3K/AKT/mTOR and MAPK in cancer cells [77]. Binding of CXCL8 to different receptors triggers signaling with distinct biological outcomes. While CXCL8/CXCR1 mainly increases tumor cell proliferation, CXCL8/CXCR2 enhances angiogenesis [78]. Additionally, CXCL16 and its receptor, CXCR6, have a stimulatory effect on cell proliferation, and their expression on tumor cells correlates with high-stage and high-grade PCa [100]. The CXCL5/CXCR5 axis has tumorigenic capacity by stimulating tumor growth and metastatic potential of PCa [101]. Additionally, CXCL5 is involved in modulating the CXCL5/STAT5/cyclin-D1 pathway that drives cell proliferation in metastatic CRPC [102]. The CXCL12/CXCR4 complex is a key factor in migration and survival of PCa cells, and indeed CXCR4 inhibitors induce apoptosis of the oncogenic cells and the suppression of cancer proliferation and vascularization [103]. The CXCL12/CXCR4 axis is fundamental to the metastatic process, and CXCR4 upregulation has been related to a more aggressive PCa phenotype and poor patient survival [104].

In BC, CXCL1 controls the epithelial-stromal interactions that facilitate tumor growth and invasion, and its overexpression is significantly associated with reduced cancer-specific and overall survival (OS) [105]. The elevated expression of CXCL12 and CXCR4 has been correlated with high grade and advanced stage of primary and recurrent BC [106]. CXCL5, found to be higher in BC tissue than in normal bladder mucosa, contributes to cell adhesion, migration, and growth via the PI3K/AKT, SNAIL, and ERK1/2 signaling pathways. The downregulation of CXCL5 impairs the migration and invasion of BC cells, whereas its upregulation can induce mitomycin C resistance via the NF-κB complex [107]. The CXCL5/CXCR2 axis favors migration and invasion in BC cell lines, increasing MMP-2/-9 levels through the PI3K/AKT pathway, and CXCL5 has been linked to tumor grade, muscle invasion, and poor OS [108]. Furthermore, urinary CXCL1 levels have been correlated with a significantly higher risk of BC relapse after transurethral resection (TUR), suggesting that CXCL1 may be used for prognostic prediction [108].

The CXCL12/CXCR4 cascade has been identified as a vital component in guiding embryological migration of primordial germ cells, and is implicated in the predisposition of testicular germ cells to cancer [109,110]. In adult human testes, CXCL12 is predominantly expressed by Sertoli cells, while CXCR4 is expressed by normal spermatogonial and meiotic germ cells, as well as most cancer cells [111]. Exposure to CXCL12 induces an invasion response in the TCam-2 seminoma cell line, but not in non-seminoma cell lines (833ke and Ntera2/D1). No evidence of CXCL12 involvement in cell proliferation or survival is seen in either seminoma or non-seminoma cell lines, suggesting that the CXCL12/CXCR4 pathway may contribute to seminoma cell migration rather than tumor growth and survival [112].

In penile cancer, preoperative serum levels of CXCL5 have been significantly linked with oncological variables such as T stage and nodal status, while survival analysis has shown a shorter disease-free survival (DFS) of patients with high serum CXCL5 levels [113]. These findings suggest a potential diagnostic and prognostic role of CXCL5 in penile cancer, although the mechanisms leading to CXCL5 upregulation remain elusive.

### 4.3. Others

MMPs are implicated in many steps of the metastatic cascade, including tumor invasion, migration, extravasation, angiogenesis, and tumor growth [114]. In RCC cells, a connection has been established between the loss of VHL function and the upregulation of membrane type-1 MMP (MT1-MMP) gene expression and protein, mediated by the HIF-2 and Sp1 transcription factors [115]. Tissue metalloproteinase inhibitors (TIMPs) strictly regulate MMP activities during tissue remodeling. In RCC patients, the increased expression of MMP-2/-9 and TIMP-1/-2 is related to high tumor grade and shortened survival [116]. The paradoxical prognostic implication of TIMP overexpression may be explained by the dual function of TIMPs, although further investigation is required. Lately, Yang et al. described the effect of a natural benzo[c]phenanthridine alkaloid (chelerythrine) on MMP modulation (i.e., the downregulation of MMP-2/-9 levels, increased TIMP-1/-2 protein expression) and its preventive role in the invasion and metastasis of androgen-independent PCa cell lines [117].

Cyclooxygenase-2 (COX-2) converts arachidonic acid into pro-inflammatory prostanoids, and its aberrant expression appears to play a part in the pathogenesis of various malignancies, including BC. However, some authors have correlated the expression of COX-2 protein to high pathological stage, vascular invasion, and lymph node metastasis [118], whereas others have surmised that COX-2 and BC relapse are inversely related [119]. Prostaglandin E2 (PGE2) is the main isoform synthesized in humans. Its activity in BC is enhanced by activation of the fibroblast growth factor receptor 1 (FGFR1), and seems to contribute to EMT through the APK/PLCγ/COX-2 pathway [120]. Urinary PGE2 levels have proved to be higher in patients with UTIs and BC than age-matched controls, and significantly decrease in successfully treated patients compared to patients with active disease [121]. More recent data have shown that PGE2 receptors EP1–4 are linked to tumor grade and cancer recurrence in non-muscle invasive bladder cancer (NMIBC) [122].

Finally, nitric oxide (NO), generated from nitric oxide synthase (NOS), is involved in the process of angiogenesis in both benign and malignant human bladder tissues [123]. BC patients exhibit significantly higher NO levels than controls, however, there is no relation between NO and tumor grade [124].

## 5. Inflammatory Cells in Genitourinary Cancers

### 5.1. Tumor-Associated Macrophages (TAMs)

Macrophages arise from immature precursors released into the bloodstream from the bone marrow, and migrate to peripheral tissues where they differ in response to specific microenvironmental signals [125]. TAMs are the principal inflammatory component of the stroma of many tumors and affect different aspects of neoplastic tissue (Figure 2).

Macrophages are known to contribute to metastasis by priming the pre-metastatic site and enabling tumor cell extravasation and survival. They are classified into proinflammatory (M1) and anti-inflammatory (M2) macrophages. The association between high TAM infiltration and poor prognosis in several cancer types suggests their role in favoring tumor growth [126], however, high TAM density has also been correlated with longer survival [127,128,129].

TAMs often observed in RCCs exhibit a high expression of the enzyme 15-lipoxygenase (LOX), which increases arachidonic acid metabolism by stimulating production of hydroxyheicosatetraenoic acids (HETEs). The upregulation of LOX-HETE affects the immune function of TAMs that secrete immunosuppressive IL-10 and are considered to be regulatory macrophages. Their presence appears to negatively affect the prognosis and efficacy of tumor vaccines and immunotherapy in RCC [130,131].

In T24 human BC cell line, pro-inflammatory macrophages initiate cell invasion and activate the PI3K/AKT signaling pathway [132]. Employing an orthotopic urinary BC model, Yang et al. stated that infiltrated TAMs can assist tumor-induced lymphangiogenesis by the paracrine secretion of VEGF-C/D [133]. Indeed, TAM depletion by clodronate liposome leads to significant inhibition of both lymphangiogenesis and lymphatic metastasis. A high number of TAMs has been correlated with greater demand for cystectomy, distant tumor spread, and shorter 5-year survival [134]. Marked CD68+ TAM infiltration following intravesical Bacillus Calmette-Guérin (BCG) immunotherapy appears to be associated with worse response to treatment and shorter relapse-free survival, however, the use of CD68+ cell density as a biomarker of tumor-induced inflammatory response necessitates further validation [135,136].

Several studies claim that TAM infiltration is a predictor for PSA failure or PCa progression after androgen deprivation therapy (ADT), with TAM < 28/HPF relating to better ADT response [137]. TAM count has also been inversely correlated with DFS after radical prostatectomy (RP), although increased TAM infiltration is not a predictor of biochemical recurrence after surgery [137]. TAM recruitment may contribute to PCa progression as well as tumor survival, angiogenesis and invasion, through the secretion of growth factors, inflammatory cytokines and chemokines, partially explaining why patients with low levels of TAMs better respond to ADT [138].

In penile cancer, TAMs are believed to participate in the formation of pre-metastatic niches that may act as a tumor survival mechanism against chemotherapy [139,140]. High TAM density has also been significantly associated with improved cancer-specific survival and lower the risk of regional recurrence [141]. The majority of these cells prove to be CD68+CD163+, indicative of M2-polarization [142].

### 5.2. Myeloid-Derived Suppressor Cells (MDSCs)

Myeloid-derived suppressor cells (MDSCs) constitute a heterogeneous cell population, morphologically resembling immature granulocytes, monocytes and dendritic cells (DCs), characterized by strong immunosuppressive and angiogenic activity [143]. The ability of these cells to suppress various types of immune response appears to have originated as protection from extensive tissue damage through unresolved inflammation. MDSCs are involved in the regulation of several diseases including infections, autoimmune diseases, trauma, and cancer. These cells inhibit T cell-mediated immunity with different mechanisms: (1) the expression of enzymes (e.g., arginase-I) interfering with the availability of amino acids (e.g., L-arginine, L-cysteine), necessary for lymphocyte activation; (2) the production of reactive oxygen species (ROS) and/or inducible NOS with consequent reduction of cytotoxic T lymphocyte activity and IFN-γ synthesis; (3) direct inhibition through regulatory T lymphocyte (Treg) expansion; (4) the reduction of L-selectin expression on naïve T lymphocytes, thus decreasing their capacity to interact with antigen-presenting cells in peripheral lymph nodes [86,143,144].

In patients with RCC as well as with other cancers, granulocytic (G) MDSCs (CD33+, HLADR−, CD15+, CD14− dominate over monocytic (M) MDSCs (CD33+, HLADR−, CD15−, CD14+), as seen in mouse models [145,146]. The high pretreatment levels of M-MDSCs and G-MDSCs have been correlated with reduced OS in metastatic RCC patients [147].

In bladder tumor xenograft models, MDSCs have proved to contribute to urothelial carcinoma progression. Eruslanov et al. identified PGE2 secreted by cancer cells as a promoter of myeloid progenitor differentiation to MDSCs, suggesting that enhanced cancer-associated inflammation and deregulated PGE2 metabolism can induce an immunosuppressive pro-tumoral phenotype myeloid cells [148]. Moreover, several studies have indicated that MDSC number correlates to advanced clinical stage and poor oncologic outcome in BC patients. Fridlender et al. demonstrated that cisplatin and gemcitabine administered after viral immunotherapy decrease the density of immunosuppressive cells including MDSCs, markedly boosting anti-tumor efficacy in multiple tumor animal models [149]. Based on this evidence, ongoing trials are investigating systemic therapies with agents known to modulate MDSCs in the metastatic and salvage settings of BC.

A larger number of circulating MDSCs has been found in PCa patients than in healthy individuals, with a significant reduction achieved following prostatectomy [150,151]. Additionally, M-MDSCs have been associated with well-established negative PCa prognostic markers, such as high levels of lactate dehydrogenase (LDH) and PSA. Greater M-MDSC density before treatment also appear to correlate with a shorter median OS [152].

Elevated levels of M-MDSCs (CD14+, HLA−DR−) prior to treatment have been linked to lower OS in CRPC patients undergoing combined prostate GVAX/ipilimumab immunotherapy [153]. Furthermore, a study on metastatic CRPC patients, randomized to receive personalized peptide vaccination (PPV) plus herbal medicines or PPV alone, showed that combination therapy stabilizes the frequency of M-MDSCs during treatment, whereas the frequency of average M-MDSC significantly increases in the PPV alone arm [154]. In metastatic CRPC, low-dose cyclophosphamide plus PPV has been associated with a decrease in M-MDSC levels and OS improvement [155]. In other studies, however, the granulocytic fraction has been claimed to be the main influencer of prognosis. Chi et al. identified G-MDSCs as the major subset in PCa, linking them to increased serum IL-8 and IL-6 levels, as well as poorer OS [156]. This finding was confirmed by Hossain et al., who observed a correlation between cancer stage and G-MDSC levels [157]. Hence, further investigations are warranted to clarify the predictive/prognostic value of the two MDSC subpopulations in PCa.

### 5.3. Tumor Infiltrating Lymphocytes (TILs)

Tumor infiltrating lymphocytes (TILs) are a selected population of T cells with higher specific immunological reactivity against tumor cells, compared with other tumor-infiltrating immune cells, such as macrophages, DCs, and mast cells. TILs are the major component of the tumor microenvironment and consist of CD3+ CD4+ (helper) and CD3+ CD8+ (cytotoxic) T cells. The role of TILs as potential prognostic and predictive biomarkers of tumor immune response has been extensively investigated, although results are not yet conclusive.

The immune system plays a critical role in the development and regulation of neoplastic kidney disease, and RCCs are often infiltrated by high levels of TILs. Although increased TILs have been correlated with better prognosis in several solid tumors, data for RCC are controversial [158]. Some authors have found a negative correlation between high CD8+ TIL density and clinical outcomes, possibly due to increased TILs often being associated with higher tumor grade [159]. Furthermore, in RCCs with high CD8+ TIL density, the increased expression of co-stimulatory checkpoints such as PD-L1 leads to negative immunomodulatory effects. Recently, an “immunoregulatory” TIL phenotype (CD8+ PD-1+ TIM-3+ LAG-3+) has been identified in RCC. The presence of these TILs and high PD-L1 expression in the immune infiltrate have been associated with aggressive histological features and high relapse rate following primary surgery [160]. Several authors have evaluated the predictive value of TILs for clinical outcome and treatment response. Siddiqui et al. reported a shorter survival in patients with FOXP3+ Tregs [161], while Granier et al. showed a higher risk of relapse and worse OS in patients with intra-tumoral CD8+ T lymphocytes co-expressing specific inhibitory receptors (i.e., PD-1 and TIM-3) [162].

In the BC microenvironment, TILs can suppress as well as promote tumor growth [163]. Some authors have demonstrated a correlation between a high number of CD3+ lymphocytes and better OS in pT1–4 BC patients undergoing radical cystectomy (RC) [164]. Sharma et al. confirmed this correlation in MIBC patients, but not in NMIBC patients treated with TUR or RC [165]. A high number of TILs at the time of TUR was also associated with higher relapse rate [166].

A higher proportion of CD4+ Th17 cells has been found in the tumor tissue than in the peripheral blood of BC patients. Although the role of Th17 cells in tumor development remains largely unknown, the balance between Th17 and Treg cells may be involved in urothelial carcinogenesis and become a therapeutic target in invasive disease [167].

Unlike in other cancers (melanoma, lung, breast, pancreatic, kidney, and bladder carcinoma), the search for TILs has met with little success in PCa, regarded as a “cold” tumor with poor T cell infiltration and high numbers of immunosuppressive Tregs and MDSCs. Some authors have reported the presence of non-functional TILs in PCa patients and the failure to stimulate them [168]. More recently, Yunger et al. have shown that isolated prostate-TILs with specific anticancer functions can be expanded and reactivated, thus supporting the development of TIL adoptive cell therapy in PCa [169].

In seminomas, CD3+ cells are mainly TILs with memory T cells, while TAMs, B cells and plasma cells are less represented. With regard to in situ seminomas, the infiltrate is predominantly composed of CD8+ and CD4+ T cells, followed by B cells, macrophages, NK and DC cells [170]. Despite the extensive lymphocytic infiltrate in seminomas, the measured CD8+ T cell activity remains low. Parker et al. revealed a borderline role of TILs in predicting the risk of relapse in a cohort of 150 men with stage I testicular seminoma subjected to orchiectomy [171]. Along with age < 33 years, tumor diameter > 6 cm, lymphatic or vascular invasion, and stromal invasion of the rete testis, a low number of TILs was associated with a higher risk of relapse.

The tumor microenvironment in penile cancer is extremely complex. Particularly in HPV-related penile SCC, the amount of tumor-associated immune cells is increased [172]. Besides T cells, memory B cells residing in the penis also appear to act as the drivers of adaptive immunity for humoral response [173]. Several studies have established that CD20+ B cells improve prognosis in various cancer types, however, others have shown a neutral effect in the invasive SCC of the penis [174,175]. Hladek et al. assessed the tumor-associated immune cell infiltrate density (ICID) with antibodies against CD3, CD8, and CD20 [176]. In contrast to the meta-analysis data showing CD3+ and CD8+ cells to be associated with better prognosis [177], these authors found no significant effect of CD3+ cells on either OS or lymph node status. Considerable interest has been focused on identifying the main immune infiltration pattern in penile SCC. Chahoud et al. analyzed the cytotoxic T cell population sub-type (CD3+, CD8+, PD-1+), showing a significant association between high densities of stromal cytotoxic T cells and worse median OS [141]. Other authors have suggested a relationship between low stromal CD8+ T cells and lymph node metastases [142], while Vassallo et al. reported worse DFS probability in tumors with high levels of FOXP3+ Treg cells [175].

## 6. Inflammation as a Therapeutic Target in Genitourinary Cancers

The association between chronic inflammation and cancer has urged researchers to target inflammation in cancer care. Such targets include COX, NF-kB, cytokines/chemokines and their receptors, FGF and its receptor, as well as VEGF. In several neoplasms (i.e., pancreatic, prostate, cervical, breast, lung, and colon), the overexpression of COX has been linked to enhanced angiogenesis, a crucial step in tumor invasion and metastasis, as well as increased apoptosis resistance [178,179].

Non-steroidal anti-inflammatory drugs (NSAIDs) are COX competitive inhibitors that have been employed for cancer therapy and prevention [180]. The role of the COX-2/PGE2 signaling pathway in colorectal tumorigenesis is established, but remains to be defined in other tumor types [181]. Dhawan et al. tested a short-term pre-surgical administration of celecoxib (selective COX-2 inhibitor) in patients with invasive BC selected for cystectomy, which resulted in increased tumor cell apoptosis [182]. Conversely, in a randomized phase IIb/III trial designed for NMIBC, Sabichi et al. demonstrated only marginal reduction in metachronous recurrence after the administration of celecoxib versus placebo [183].

The IL-15 super-agonist ALT-803 consists of two molecules of IL-15 N-to-D substituted at position 72 linked to a dimeric IL-15 receptor alpha (Rα) “sushi” domain/IgG1 Fc fusion protein. In a phase I trial in BCG-naïve patients with NMIBC, combination therapy with ALT-803 and BCG exhibited antitumor activity with a favorable safety profile, with all patients disease-free at 24 months [184]. Based on these data, a single-arm phase II study assessing ALT-803 plus BCG in patients who have failed prior BCG therapy is ongoing (NCT03022825).

ALT-801 is a recombinant humanized T cell receptor (TCR)-IL-2 fusion protein, which fuels NK cell and T cell cytotoxic immune responses against p53-expressing tumor cells. In a phase I trial, intravenous infusion of ALT-801 and gemcitabine has promised durable clinical activity in BCG-resistant NMIBC patients [185].

Over the last few years, novel strategies targeting pro-inflammatory immunity in the treatment of PCa patients have emerged [186]. Although siltuximab (IL-6 inhibitor) impedes castration-resistant progression in androgen-dependent PCa xenograft models [187], no clinical benefit has been confirmed in two phase II clinical trials in metastatic CRPC patients (NCT00433446, NCT00385827). Nevertheless, due to the decrease in phosphorylated STAT3 and mitogen-activated protein kinases in RP specimens, siltuximab may be effective early in disease progression [188].

Niclosamide, an oral anti-helminthic drug, suppresses macrophage-induced inflammation via STAT3 and/or NF-κB signaling and has been identified as a potent androgen receptor splice variant 7 (AR-V7) inhibitor in PCa cells [189]. Clinical trials are underway to evaluate the impact of niclosamide administered with enzalutamide or abiraterone in metastatic CRPC patients (NCT03123978, NCT02807805).

Pexidartinib is a CSF-1 receptor inhibitor found to reduce TAM pro-tumorigenic properties in mouse xenograft models. The addition of pexidartinib to docetaxel has shown to increase therapeutic efficacy in CRPC [190], while data are awaited on pexidartinib together with radiotherapy and ADT in patients with localized disease (NCT02472275).

In a VCaP xenograft model, the use of CXCL2 monoclonal antibody, carlumab (CNTO 888), reduced TAM infiltration while decreasing tumor growth [191]. In combination with docetaxel, carlumab significantly lowered tumor burden versus docetaxel alone in a mouse xenograft model of human PCa [192]. Single agent carlumab, however, showed no anti-tumor activity in metastatic CRPC patients (NCT00992186).

Tasquinimod oral inhibitor of S100A9, a key cell surface regulator of MDSC function, has been assessed in a randomized phase III trial versus placebo in chemotherapy-naive men with hormone refractory metastatic PCa. Tasquinimod significantly improved radiographic PFS compared with placebo, although no OS benefit was seen (NCT01234311).

Bruton tyrosine kinase (BTK) plays a major driver of B cell development as well as T2 TAM polarization, and potentially inhibits solid tumor progression [193,194]. The safety and efficacy of ibrutinib is currently being investigated in localized PCa (NCT02643667).

The enzyme indoleamine 2,3-dioxygenase (IDO), an immunomodulatory enzyme involved in tumor immune escape, restrains CD8+ effector T cells and NK cells, activating Tregs and MDSCs [195,196]. A phase II study has been conducted with indoximod, an IDO1 pathway inhibitor, versus placebo after the completion of standard of care sipuleucel-T in metastatic CRPC patients (NCT01560923). Oral indoximod administration twice daily for 6 months is well tolerated, substantially improving radiographic and clinical progression.

Anakinra, an IL-1 receptor antagonist, has proved to be an effective therapeutic option in PCa [197]. Besides IL-1 targeting, it reduces the level of IL-6, which is among the most common cytokines associated with PCa progression and metastasis. Anti-IL-6 antibodies halt tumor growth in vitro and in vivo, however, clinical application of anti-IL-6 therapies does not appear to lengthen survival in metastatic PCa patients.

Recently, simvastatin, all-trans retinoic acid (ATRA), and PI3K/AKT pathway inhibitors have given encouraging results in prolonging RCC patient survival [14]. Simvastatin, a cholesterol-lowering drug, inhibits phosphorylation of AKT, mTOR and ERK, thus hampering the proliferation and migration of RCC cells. Furthermore, simvastatin also has antitumor effects preventing the IL-6-induced phosphorylation of JAK2 and STAT3 [198]. ATRA, the active metabolite of vitamin A, reduces cell proliferation and alters gene expression through RAR/RXR and PI3K/AKT pathways, prompting cell differentiation and apoptosis [199]. A powerful, competitive and reversible inhibitor of the ATP binding site of PI3K, LY294002, has also been proposed as an effective agent in the treatment for RCC patients [200]. In addition, the use of antibodies against 15-lipoxy-genase 2/15(S)-hydroxyeicosatetraenoic (15-LOX2) affects the arachidonic acid metabolism of TAMs isolated from RCC, thus reducing local immunosuppression and tumor escape [201].

Results of the enrolled studies are detailed in Table 1.

## 7. Conclusions

Inflammation is closely linked to cancer and plays a pivotal role in tumor development and progression. A microenvironment rich in inflammatory cells, growth factors, and DNA damaging agents, induces tissue injury and the accumulation of mutations in epithelial cells, enhancing their growth. The mutated cells, in turn, synthesize cytokines and recruit inflammatory cells, thus establishing an inflammatory tumor microenvironment, which participates in angiogenesis, migration, and metastasis. Since the expression of inflammatory mediators is higher in tumors than in normal tissues, the use of anti-inflammatory drugs, alone or together with chemotherapy, could provide a valid contribution to preventing and treating cancer.

## Figures and Tables

**Figure 1 ijms-24-00866-f001:**
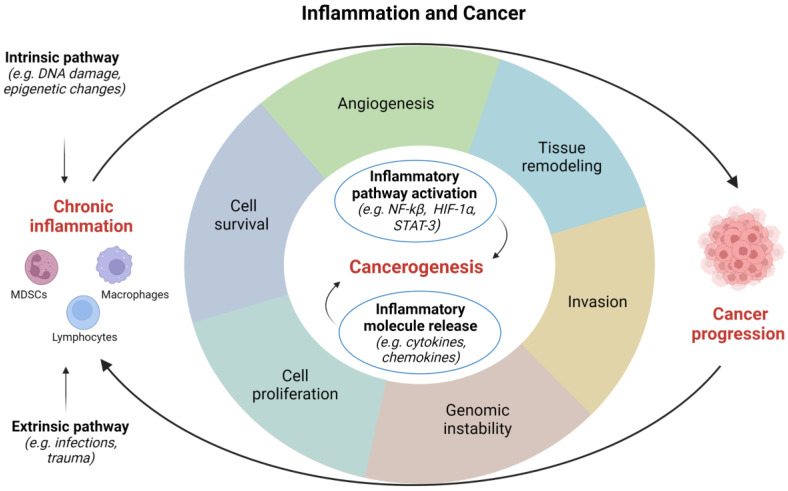
Correlations between inflammation and cancer. *HIF*, hypoxia inducible factor; *NF*, nuclear factor; *STAT*, signal transducer and activator of transcription. Image created with BioRender.com (accessed on 1 December 2022).

**Figure 2 ijms-24-00866-f002:**
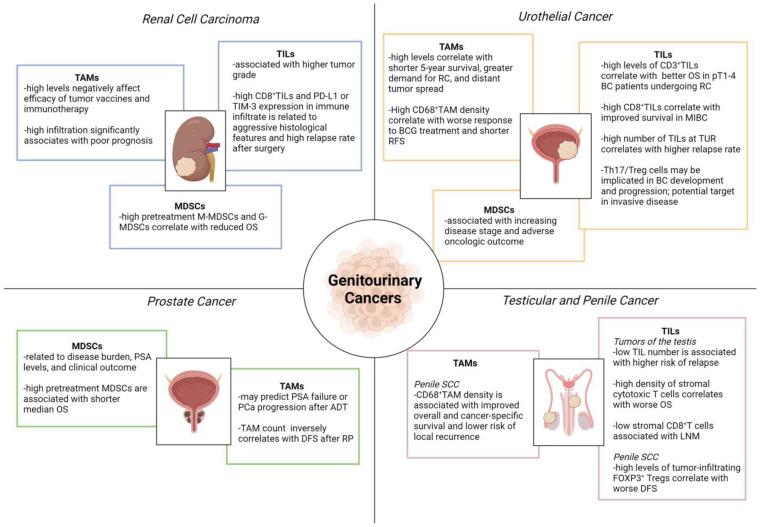
Role of tumor-infiltrating immune cells in genitourinary cancers. ADT, androgen deprivation therapy; BC, bladder cancer; BCG, Bacillus Calmette-Guérin; DFS, disease-free survival; LNM, lymph node metastasis; MDSCs, myeloid-derived suppressor cells; G-MDSCs, granulocytic myeloid-derived suppressor cells; M-MDSCs, monocytic myeloid-derived suppressor cells; MIBC, muscle-invasive bladder cancer; OS, overall survival; PD-L1, programmed death-ligand 1; PSA, prostate-specific antigen; RC, radical cystectomy; RFS, relapse-free survival; RP, radical prostatectomy; SCC, squamous cell cancer; TAMs, tumor-associated macrophages; TILs, tumor-infiltrating lymphocytes; TIM-3, T cell immunoglobulin and mucin-domain containing-3; TUR, transurethral resection; UC, urothelial cancer. Image created with BioRender.com (accessed on 1 December 2022).

**Table 1 ijms-24-00866-t001:** Studies evaluating inflammation as a therapeutic target in genitourinary cancers.

Study	Type of Study	No. of Patients	Population	Experimental Drugs	Primary Endpoint	Results	AEs
Dhawan et al. [182]	Prospective study	13	Invasive BC	Celecoxib	Tumor cell apoptosis	Apoptosis more frequent in celecoxib vs. controls (*p* < 0.04)	Mild symptoms (e.g., fatigue, stomach and back pain) in 2 cases
Sabichi et al. [183]	Phase II	146	High-risk NMIBC	Celecoxib	TTR	Average TTR delayed from 1.74 to 2.86 yrs	449 AEs, including 60 G ≥ 3 events; no deaths
Rosser et al. [184]	Phase I	9	BCG-naïve NMIBC	ALT-803 (IL-15 super-agonist) plus BCG	Antitumor activity	All pts disease-free at 24 mos	Hematuria and urinary tract pain related to BCG in all pts; hypertension (G3) in 1 case; no G4 AEs or deaths
NCT03022825	Phase II	200	BCG unresponsive high-grade NMIBC	ALT-803 plus BCG	CR; DFR	NA	NA
NCT00433446	Phase II	62	mCRPC	Siltuximab (IL-6 inhibitor)	PSA response	PSA RR 3.8%	DIC and cerebral ischemia (G4) in 1 case; elevated AST/ALT; gastritis; hematologic toxicity (G3)
NCT00385827	Phase II	106	HRPC	Siltuximab + mitoxantrone/prednisone	AEs; SAEs	NA	NA
NCT03123978	Phase I	12	mCRPC	Niclosamide plus enzalutamide	AEs; RP2D	NA	NA
NCT02807805	Phase II	37	CRPC	Abiraterone plus niclosamide	PSA response	NA	NA
NCT01560923	Phase II	63	mCRPC	Indoximod after sipuleucel-T	Immune response to sipuleucel-T	No difference in PSA progression; higher median rPFS with indoximod (10.3 vs. 4.1 mos, *p* = 0.011)	No significant difference between the two arms
NCT02472275	Phase I	8	Unfavorable risk PCa	Pexidartinib	AEs; MTD	NA	NA
NCT02643667	Phase II	27	Localized PCa	Ibrutinib	DLT	NA	NA
NCT01234311	Phase III	1245	mCRPC	Tasquinimod	PFS	Higher median rPFS with tasquinimod (7.0 vs. 4.4 mos, *p* < 0.001)	G ≥ 3 AEs 42.8%
NCT00992186	Phase II	46	mCRPC	CNTO 888 (mAb anti-CCL2)	Composite response	1 SD > 6 mos; 14 SD ≥ 3 mos; median OS 10.2 mos; no PSA or radiological response	G ≥ 3 AEs 67%; SAEs including pneumonia, spinal cord compression, back pain

## Data Availability

Not applicable.

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
