# Peer review of "Inflammation in Urological Malignancies: The Silent Killer"

_ijms, 2023, doi:10.3390/ijms24010866_

Round 1

Reviewer 1 Report

IJMS-2005985

Inflammation in Urological Malignancies: The Silent Killer

This review article summarizes molecular and cellular aspects of inflammation-driven genitourinary cancers. Pro-inflammatory infectious agents (e.g., bacterial and viral infections), mediators, signaling pathways, and cell types contributing to cancer development and progression are described. Moreover, recent anti-inflammatory strategies as therapeutic approaches to treat cancers (including their pros and cons) are discussed.

Text and figures are straightforward and clear. The article adequately covers the topic, reflects the relevant parts of the literature (including the latest reports), and provides a broad and informative overview for the reader. However, some questions have to be addressed and additional information has to be provided.

1.      Lines 88-89: Please provide more information on the connection between inflammatory response and genomic instability. What is the proposed mechanism by which Schistosoma haematobium induces K-RAS mutations?

2.      Lines 105-106: Please provide a few more details on the underlying mechanisms by which toxin-mediated oncogenesis affects genomic instability, resistance to cell death, and proliferative signaling.

3.      Lines 114-116, 144-145: Is there a difference in the respective patient populations showing an association of HPV or spinal cord injury with BC when compared to populations without an association?

4.      Lines 151-152: Please provide more background information on the modulation of the immune response and the support of prostatitis by estrogens.

5.      In chapter 3.1-3.7, the factors mentioned are not described in detail. The authors should refer to review articles providing detailed background information on the molecular/functional characteristics of the respective factors (VEGF, VHL, ...).

6.      Line 176: What are the HIF-independent functions of pVHL?

7.      Page 193: Please describe in which way S6K1 and eIF-4E contribute to HIF accumulation.

8.      Line 195: The authors state that “the PI3K/AKT/mTOR signaling pathway is activated by chemokine synthesis”. Is it correct that the activation is driven by the process of chemokine synthesis as such (and not by the synthesized chemokines)?

9.      Line 218:  Please provide some examples for the “… cytokine receptors and growth factors …” mentioned.

10.   Chapter 3.6: Which specific NF-κB subunits are involved?

11.   Line 289-290: “However, while some in vitro and in vivo studies demonstrate a reduction in IL-6-associated cell proliferation, migration and invasion, …”. In response to what?

12.   Line 306-308: “The CXCL12/CXCR4 axis induce a cascade of signals that internalizes the complex in the nucleus, favoring cellular growth, migration, and invasion”. Please describe the respective mechanism. Does the CXCL12/CXCR4 complex act as a transcription factor?

13.   Chapter 4.3: Is there any information available on modulated MMP activity in genitourinary cancers?

14.   Line 365-366: Since both MMP-2/-9 and TIMP-1/-2 are increased inn RCC: Is the ratio among proteases and inhibitors dysregulated (favoring an excess of active MMP)?

15.   477-480: Is the frequency of M-MDSCs higher in the controls?

16.   Chapter 6: Please comment on adverse effects of the therapeutic approaches described (if any).

Reviewer 2 Report

This is an excellent work focus on a crucial scientific argument. The role of inflammation in urological cancers could be a wide research area.

I definitely suggest this manuscript to be published, but I would like to see some tables to be added, including the variety of enrolled studies, their results and limitations.

Round 2

Reviewer 1 Report

ijms-2005985

Inflammation in Urological Malignancies: The Silent Killer

The manuscript provides a revised version of the manuscript “Inflammation in Urological Malignancies: The Silent Killer”. The manuscript has been improved considerably and my comments have been adequately addressed.